# A LAMP at the end of the tunnel: A rapid, field deployable assay for the kauri dieback pathogen, *Phytophthora agathidicida*

Richard C. Winkworth[1,2]*, Briana C. W. Nelson[2], Stanley E. Bellgard[3], Chantal M. Probst[3], Patricia A. McLenachan[2], Peter J. Lockhart[1,2]

1 Bio-Protection Research Centre, Massey University, Palmerston North, New Zealand, 2 School of Fundamental Sciences, Massey University, Palmerston North, New Zealand, 3 Manaaki Whenua–Landcare Research, Auckland, New Zealand

* r.c.winkworth@massey.ac.nz

**Data Availability Statement:** All relevant data are available from GenBank using the accession

## Abstract

The root rot causing oomycete, *Phytophthora agathidicida*, threatens the long-term survival of the iconic New Zealand kauri. Currently, testing for this pathogen involves an extended soil bioassay that takes 14–20 days and requires specialised staff, consumables, and infrastructure. Here we describe a loop-mediated isothermal amplification (LAMP) assay for the detection of *P. agathidicida* that targets a portion of the mitochondrial apocytochrome b coding sequence. This assay has high specificity and sensitivity; it did not cross react with a range of other *Phytophthora* isolates and detected as little as 1 fg of total *P. agathidicida* DNA or 116 copies of the target locus. Assay performance was further investigated by testing plant tissue baits from flooded soil samples using both the extended soil bioassay and LAMP testing of DNA extracted from baits. In these comparisons, *P. agathidicida* was detected more frequently using the LAMP test. In addition to greater sensitivity, by removing the need for culturing, the hybrid baiting plus LAMP approach is more cost effective than the extended soil bioassay and, importantly, does not require a centralised laboratory facility with specialised staff, consumables, and equipment. Such testing will allow us to address outstanding questions about *P. agathidicida*. For example, the hybrid approach could enable monitoring of the pathogen beyond areas with visible disease symptoms, allow direct evaluation of rates and patterns of spread, and allow the effectiveness of disease control to be evaluated. The hybrid LAMP bioassay also has the potential to empower local communities to evaluate the pathogen status of local kauri stands, providing information for disease management and conservation initiatives.

## Introduction

The long-term survival of kauri, *Agathis australis* (D.Don) Loudon (Araucariaceae), is threatened by the oomycete *Phytophthora agathidicida* B.S. Weir, Beever, Pennycook & Bellgard (Peronosporaceae) [1]. This soil-borne pathogen causes a root rot that results in yellowing of

numbers listed in Supporting Information files S1 Table and S3 Table.

**Funding:** P.J.L. and R.C.W., BPRC_MU_2016_1, BioProtection Research Centre (https:// bioprotection.org.nz). The funders had no role in study design, data collection and analysis, decision to publish, or preparation of the manuscript. P.J.L. and R.C.W., MAU1702, New Zealand Ministry of Business, Innovation and Employment Catalyst: Seeding Fund (https://royalsociety.org.nz/what-we-do/funds-and-opportunities/catalyst-fund/catalyst-seeding/). The funders had no role in study design, data collection and analysis, decision to publish, or preparation of the manuscript. R.C.W. and P.J.L., NP94607, Massey University (https://massey.ac. nz). The funders had no role in study design, data collection and analysis, decision to publish, or preparation of the manuscript. S.E.B, C09X1704, New Zealand Ministry of Business, Innovation and Employment Strategic Science Investment Fund (https://www.mbie.govt.nz/science-and-technology/science-and-innovation/funding-information-and-opportunities/investment-funds/strategic-science-investment-fund/). The funders had no role in study design, data collection and analysis, decision to publish, or preparation of the manuscript.

**Competing interests:** The authors have declared that no competing interests exist.

the foliage, bleeding cankers on the lower trunk, thinning of the canopy and eventually in tree death. Initially reported from Great Barrier Island in the early 1970's [2], since the late 1990's the disease has spread rapidly across northern New Zealand [3].

Broad phylogenies for the Peronosporaceae [e.g., 4–7] have identified strongly supported sub-clades. For example, Bourret et al. [7] described total of 16 sub-clades, the majority of which contain at least one *Phytophthora* species. The phylogenetic analysis of Weir et al. [8] placed *P. agathidicida*, along with three other formally recognized species, within Clade 5. Although *P. agathidicida* is currently the only clade 5 species reported from New Zealand, it is not the only *Phytophthora* species associated with kauri forests. A further five *Phytophthora* species have been reported from kauri forest soils. Specifically, *P. chlamydospora* Brasier and Hansen (Clade 6), *P. cinnamomi* Rands (Clade 7), *P. cryptogea* Pethybr. & Laff. (Clade 8), *P. kernoviae* Brasier (Clade 10), and *P. nicotianae* Breda de Haan (Clade 1) [1, 3]. Other oomycetes, including several *Pythium* (Pythiaceae) species, have also been reported from kauri forest soils [3].

Methods for the isolation and identification of *Phytophthora* typically involve recovery from infected host root material [9, 10] or from soil samples using plant tissue fragments (e.g., detached leaves) as bait for zoospores [11, 12]. Currently, the preferred method of testing for *P. agathidicida* is an extended soil bioassay [13]. Briefly, soil samples are first air dried (2–8 days), then moist incubated (4 days), and finally flooded with water and baited (2 days). At this point the baits are surface sterilised, plated onto *Phytophthora*-selective media, and incubated at 18°C (6 days). *Phytophthora agathidicida* is identified from the resulting cultures using morphology or molecular diagnostics both of which require trained laboratory staff. The former is typically based upon the size and reflection of the antheridium [14] while an RT-PCR assay targeting the internal transcribed spacer (ITS) regions of the ribosomal DNA [15] is routinely used for identification of cultured *P. agathidicida* isolates.

Increasing the immediacy of results from diagnostic testing can substantially improve outcomes in terms of disease control. Historically, shortening the time needed to diagnose a disease has involved the introduction of new laboratory protocols or tools. For example, culture-based methods have in some cases been replaced by molecular diagnostics. More recently, diagnostic approaches that can be conducted onsite, thereby reducing reliance on centralised laboratory facilities, have become an increasingly important means of enhancing the immediacy of results [16, 17]. The emergence of new amplification technologies is enabling the development of rapid, field-deployable approaches to genetic diagnostics [18]. In particular, loop-mediated isothermal amplification (LAMP), an approach that combines rapid amplification with high specificity and sensitivity, is rapidly becoming an important diagnostic tool, especially for point of care applications. There are now numerous examples of LAMP tests for applications in human medicine [e.g., 19, 20], agriculture [e.g., 21, 22], and animal health [e.g., 23, 24]. LAMP has several features that make it particularly well suited to non-laboratory applications. First, the DNA polymerases used to catalyse the LAMP reaction have strand-displacement activity. Therefore, unlike PCR-based diagnostics, LAMP assays may be carried out using relatively simple equipment [25]. Second, the LAMP polymerases are less sensitive to inhibitors than those used for PCR reactions allowing simpler methods of DNA isolation to be used [26]. Finally, turbidity or colorimetry can be used for end-point detection of LAMP products, again reducing reliance on sophisticated equipment [26].

The cost of the extended soil bioassay together with the requirement for specialised staff and infrastructure means that, typically, this assay is conducted only after the appearance of physical disease symptoms. Such restricted testing severely limits our basic understanding of *P. agathidicida* and our ability to make informed management decisions at local, regional and national scales. For example, we are yet to characterise the distribution of the pathogen, how

fast it is spreading, or the efficacy of interventions (e.g., track closures) aimed at disease control. Addressing these knowledge gaps requires ongoing, active monitoring of both diseased and healthy sites across the distribution of kauri. This cannot be achieved using the existing test. Instead, a reliable and rapid assay for *P. agathidicida* that is both cost effective and robust enough to be deployed outside of a laboratory is needed. Beyond increasing existing capacity, such testing has the potential to enable individual landowners and community groups to evaluate pathogen status in their area and thereby engage in an informed way with regional and national initiatives.

LAMP assays have already been reported for the detection of several *Phytophthora* species. These include tests for *P. capsici* Leonian [27], *P. cinnamomi* [28], *P. infestans* (Mont.) de Bary [29, 30], *P. melonis* Katsura [31], *P. nicotianae* [32], *P. ramorum* Werres, De Cock & Man in 't Veld [22] and *P. sojae* Kaufm. & Gerd. [33]. The genetic targets of these tests include the Ras-related protein (*ypt1*) gene [i.e., 30–33] and the ITS regions [i.e., 22, 27]. As might be expected, LAMP tests for *Phytophthora* species differ in terms of both their absolute detection limits and performance relative to PCR-based assays. For example, Hansen et al. [29] and Khan et al. [30] reported *P. infestans* LAMP tests with detection limits of between 128 fg and 200 pg. The most sensitive of these, that of Khan et al. [30], was ten times more sensitive than a test based on nested PCR and at least 100 times more sensitive than either RT-PCR or conventional PCR tests for the corresponding locus.

Here we describe a hybrid bioassay for the detection of *P. agathidicida*. This combines conventional soil baiting with a highly specific and sensitive LAMP assay to directly test the plant bait tissues for the presence of the pathogen. By reducing assay cost, the time needed for pathogen detection, and reliance on centralised laboratories this approach has the potential both to overcome key limitations of the currently used extended soil bioassay and provide data that will inform our basic understanding of the disease and its management.

## Materials and methods

### Target region identification

To identify potential targets for genetic testing, publically available *Phytophthora* and *Pythium* mitochondrial genome sequences were obtained from the NCBI RefSeq database (https://www.ncbi.nlm.nih.gov). These were combined with mitochondrial genome sequences from *Phytophthora* and related taxa (e.g., *Pythium*, *Plasmopara*) assembled at Massey University. The combined collection comprised mitochondrial genomes from 25 species representing 12 of the 16 clades of *Phytophthora* and downy mildews reported by Bourret et al. [7]. This sample included all five *Phytophthora* species reported from kauri forests [1, 3], all currently recognized representatives of *Phytophthora* clade 5 [8], and two accessions of *P. agathidicida* [8]. A list of species and accession numbers for publicly available sequence data are provided in S1 Table.

To identify potential targets for LAMP assays, mitochondrial genome sequences from five *Phytophthora* clade 5 species were aligned using the MUSCLE [34] alignment tool as implemented in Geneious V9 (Biomatters, Auckland, New Zealand). To evaluate the utility of the identified loci, 0.5–1 kb sections of DNA sequence containing these regions were extracted from all 25 mitochondrial genomes and multiple sequence alignments for each constructed as before.

### LAMP primer design

LAMP primer sets were generated for potential target regions using PrimerExplorer v5 (https://primerexplorer.jp/e/). For each region, an initial search for regular primers used the *P.*

*agathidicida* sequence along with default parameter values; the GC content threshold was progressively lowered in subsequent searches until at least one regular primer set was recovered. Candidate primer sets were then compared to multiple sequence alignments for the corresponding target locus; primer sets where annealing sites did not distinguish *P. agathidicida* were not considered further. For the remaining primer sets, a search was then made for loop primers using PrimerExplorer v5 and the same GC content threshold as generated the regular primer set.

Finally, each primer set was queried against the NCBI nucleotide database (https://www.ncbi.nlm.nih.gov) using BLAST [35] to investigate non-specific annealing.

### LAMP assay optimisation

For reaction optimisation, all LAMP assays were conducted in 25 µL volumes consisting of 15 µL OptiGene Isothermal Master Mix (OptiGene Ltd., Horsham, West Sussex, England) plus primer cocktail, extracted DNA and milliQ water ($H_2O$). Volumes of the three latter components were varied depending on the reaction conditions and template concentration. Reaction sets typically included both positive (i.e., 2 ng total DNA from cultured *P. agathidicida* isolates NZFS 3128 or ICMP 18244; see S1 Table for accession details) and negative (i.e., no DNA) controls. LAMP assays were performed using a BioRanger LAMP device (Diagenetix, Inc., Honolulu, HI).

Initial optimisation of the LAMP assay evaluated three parameters. First, we investigated the impact of varying the ratio of F3/B3 to FIP/BIP primer pairs. Ratios of F3/B3 to FIP/BIP primers of 1:3, 1:4, 1:6, and 1:8 were trialled; in each case the final concentration of the F3 and B3 primers was maintained at 0.2 µM with the concentrations of FIP and BIP primers being 0.6 µM, 0.8 µM, 1.2 µM, and 1.6 µM, respectively. Second, we examined the effect of amplification temperature. LAMP assays were performed at amplification temperatures of 60˚C, 63˚C, and 65˚C. In all cases, DNA amplification was followed by enzyme denaturation at 80˚C for 5 min. Finally, the amplification time was varied from 30–90 minutes.

### LAMP assay specificity

The specificity of the *P. agathidicida* LAMP assay was evaluated using optimised reaction mixes and conditions. These tests were conducted using 2 pg total DNA from six *P. agathidicida* isolates as well as from isolates of 11 other *Phytophthora* species representing nine of the 16 clades reported by Bourret *et al.* [7]. All *Phytophthora* species reported from kauri forests and all currently recognized representatives of *Phytophthora* clade 5 were included in the test set (S1 Table). Individual reaction sets also included both positive (i.e., 2 ng total DNA from cultured *P. agathidicida* isolates NZFS 3128 or ICMP 18244) and negative (i.e., no DNA) controls.

### LAMP assay sensitivity

We first evaluated LAMP assay sensitivity in terms of total *P. agathidicida* DNA. For these tests, a ten-fold dilution series with between 1 ng and 10 ag of total DNA from cultured *P. agathidicida* isolate ICMP 18244 together with optimised reaction mixes and conditions were used. Testing of the dilution series was conducted both with and without the addition of DNA from a plant tissue commonly used when baiting for *P. agathidicida*; specifically, 2 ng total *Cedrus deodara* (Roxb.) G.Don (Himalayan cedar) DNA. Reaction sets also included both positive (i.e., 2 ng total DNA from cultured *P. agathidicida* isolates NZFS 3128 or ICMP 18244) and negative (i.e., no DNA) controls.

We also evaluated assay sensitivity in terms of target copy number. As a template for these tests, we produced a PCR fragment 799 base pairs (bp) in length. Amplifications were typically performed in 20 μL reaction volumes containing 1× EmeraldAmp GT PCR Master Mix (Takara Bio Inc., Kusatsu City, Shiga Prefecture, Japan) and 0.5 μM of each amplification primer (PTA_pcrF, 5'-CCAAACATAGCTATAACCCCACCA-3'; PTA_pcrR, 5'-GGTTTC GGTTCGTTAGCCG-3'). Thermocycling was performed using a T1 Thermocycler (Biometra GmbH, Göttingen, Germany) and with standard cycling conditions including an initial 4 min denaturation at 94˚C, then 35 cycles of 94˚C for 30 secs, 58˚C for 30 secs and 72˚C for 30 secs, with a final 5 min extension at 72˚C. Amplification products were prepared for DNA sequencing using shrimp alkaline phosphatase (ThermoFisher Scientific, Waltham, MA) and exonuclease 1 (ThermoFisher Scientific), following a manufacturer recommended protocol. Sensitivity tests using the PCR fragment as template were conducted as for total DNA.

## Comparison of standard bioassay and hybrid LAMP bioassay

A direct comparison of the extended bioassay and hybrid LAMP bioassay was performed for two sets of soil samples, one collected from sites in the Waitākere Ranges Regional Park and the other from the Waipoua Forest Sanctuary (S2 Table). These collections were made by the Healthy Trees Healthy Future (HTHF) programme following consultation with representatives of the mana whenua–Te Kawerau ā Maki (Waitākere Ranges) and Te Roroa (Waipoua)– and under permit from the New Zealand Department of Conservation (e.g., 69218-GEO). Samples from each site typically consisted of 1–2 kg of soil from the upper 15 cm of the mineral horizon in the vicinity of kauri trees displaying dieback symptoms. Subsamples of 500 g were first air dried in open plastic containers for two days, moist incubated for four and subsequently flooded with 500 mL of reverse osmosis H$_2$O. Fifteen detached *Cedrus deodara* needles were then floated on the water surface. Baits were removed after 48 h (Fig 1); ten were immediately used for the standard bioassay with the remainder frozen at -20˚C prior to DNA extraction and LAMP testing.

For the standard bioassay, cedar baits were first rinsed with reverse osmosis (RO) H$_2$O, soaked in 70% ethanol for 30 s, then rinsed again with RO H$_2$O before being dried on clean paper towels. Surface sterilised baits were then placed on *Phytophthora*-selective media [36] using sterile technique and incubated at 18˚C for 5–7 days. To identify the resulting cultures, asexual and sexual structures were examined using a Nikon ECLIPSE 80i compound light microscope (Nikon Corporation, Tokyo, Japan) with micrographs captured using a Nikon DS-Fi1 digital microscope camera head (Nikon Corporation) and processed using NIS-Elements BR (version 5.05, Nikon Corporation) [8, 37] (Fig 1).

For the hybrid LAMP bioassay total DNA was extracted from two or three frozen cedar baits using the Macherey-Nagel Plant Kit II (Macherey-Nagel GmbH & Co. KG, Düren, Germany) and the manufacturer's recommended protocol for plant material. Following extraction, the concentration of total DNA was determined for each sample using a Qubit 2.0 Fluorometer (Thermo Fisher Scientific, Waltham, MA). LAMP assays were performed using optimised reaction mixes and conditions with up to 5 ng of total bait DNA added as template. Each reaction set included both positive (i.e., 2 ng total DNA from a cultured isolate of *P. agathidicida*) and negative (i.e., milliQ H$_2$O) controls (Fig 1).

We also used PCR and sequencing of the ITS region to assess the presence of *P. agathidicida* in total bait DNA samples. Amplifications were performed in 25 μL volumes containing 1 × Platinum SuperFi PCR Master Mix (Invitrogen, Carlsbad, California, USA), 12.5 pM of amplification primer ITS_PTA_F2 [15], 12.5 pM of amplification primer ITS4 [38], and 4 ng of total bait DNA. Thermocycling consisted of an initial 30 sec denaturation at 98˚C, followed

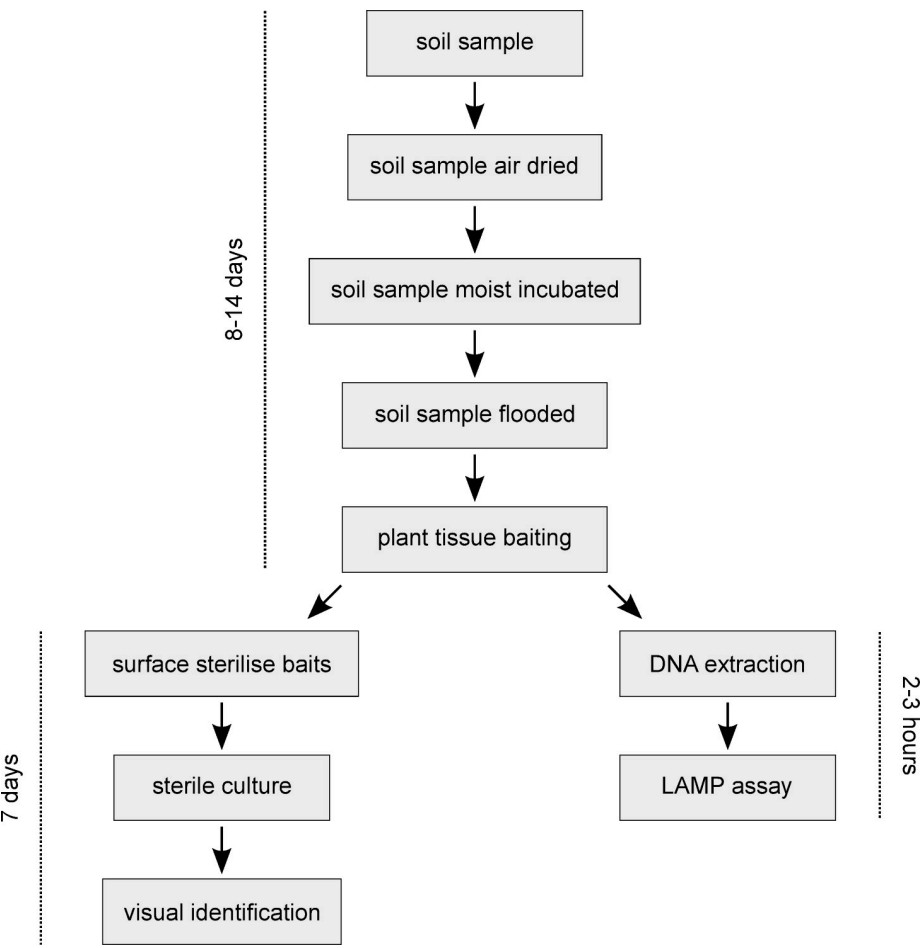

**Fig 1. Schematic diagram comparing the extended bioassay and hybrid LAMP bioassay workflows and timelines.**

by 35 cycles of 98˚C for 10 secs, 58˚C for 10 secs, and 72˚C for 30 secs with a final extension of 72˚C for 5 mins. Amplification products were purified using the Macherey-Nagel NucleoSpin Gel and PCR Clean-up (Macherey-Nagel GmbH & Co. KG) following the manufacturer's recommended protocol. Sequencing products were generated from each amplification primer using ABI PRISM BigDye Terminator Cycle Sequencing Ready Reaction Kits (Applied Biosystems, Foster City, California, USA) and run on an ABI 3730 DNA Analyzer (Applied Biosystems). Sequences for each amplification product were assembled using Geneious R9 (Biomatters) and queried against the NCBI nucleotide database (https://www.ncbi.nlm.nih.gov) using BLAST [35].

To further assess the presence of *P. agathidicida* on baits we also conducted whole genome sequencing of total bait DNA from each of the three sampling sites in the Waitākere Ranges Regional Park. Specifically, we sequenced HTHF 1014, HTHF 1020, and HTHF 1035 (S2 Table). Shotgun sequencing libraries were prepared for each DNA extraction using Illumina Nextera DNA library preparation kits (Illumina, Inc., San Diego, CA). The Massey Genome Service (Palmerston North, New Zealand) performed library preparation, paired-end DNA sequencing and quality assessment of the resulting reads. For each sample a preliminary *de novo* assembly was performed using idba_ud [39]. The resulting contigs were then compared to our collection of mitochondrial genome sequences (S1 Table) using BLAST [35]. Using the

**Table 1. Primer sequences for the *P. agathidicida* loop-mediated isothermal amplification (LAMP) assay.**

| Name | Sequence (3' to 5') |
|---|---|
| PTAF3 | TTATTTGAACCAACCTCATGT |
| PTAB3 | TGTTTTACCTTGGGGACAA |
| PTALF | TTAGTTTACATTTTACTTTTCCTTTTG |
| PTALB | CCTATTAAAGGTATTGCAGAAAATAA |
| PTAFIP | GCTGTAGATAATCCAACTTTAAATCGTTTT–GGTGTATTAATACGACCCCTAC |
| PTABIP | CCACCCCATAGCCAATCAACAATA–TTTTGGGGTGCAACTGTT |

reference assembly tool implemented in Geneious R9 (Biomatters), contigs with high similarity to the reference set were then mapped to a complete mitochondrial genome sequence of *P. agathidicida* (S1 Table), *P. cinnamomi* (S1 Table), and *Pythium ultimum* Trow [40]. Assemblies were subsequently checked by eye; contigs were removed from an assembly if similarity to another reference genome was higher.

# Results

## Target region identification and primer design

A comparison of mitochondrial genome sequences for members of *Phytophthora* clade 5 suggested several potential targets for a LAMP assay specific to *P. agathidicida*. Seven sets of LAMP primers, each consisting of between four and six primers, were designed using Primer-Explorer v5. However, in initial trials all but one of these primer sets failed to discriminate *P. agathidicida* from other Clade 5 species. We assume that although there are sequence level differences between *P. agathidicida* and other Peronosporaceae at these loci, the overall number and/or distribution of these differences is not sufficient to prevent amplification in non-target species. The exception was a set of six primers targeting a 227 nucleotide long section of the apocytochrome b (*cob*) coding sequence spanning from nucleotide position 392 to position 617 (Table 1; Fig 2). Of the initial primer sets, only this one was tested further.

## LAMP assay optimisation

Using 2 ng of total DNA from cultured *P. agathidicida* isolates NZFS 3128 or ICMP 18244 as template, the *P. agathidicida* LAMP assay gave similar results across a range of reaction conditions. Specifically, amplification was observed for all examined ratios of external to internal primers (i.e., 1:3, 1:4, 1:6 and 1:8), amplification temperatures (i.e., 60°C, 63°C, and 65°C), and amplification times (i.e., 30 min, 45 min, 60 min, and 90 min). Conversely, no amplification was observed under these same reaction conditions for controls containing no DNA.

For subsequent analyses, a 1:3 ratio of external to internal primers, an amplification temperature of 63°C, and an amplification time of 45 min were used.

## LAMP assay specificity and sensitivity

Using optimised reaction mixes and conditions, we consistently recovered amplification products from tested *P. agathidicida* isolates (Table 2). In all cases, amplification was detected using real-time fluorescence (e.g., Fig 3, panel B curves B and C) and agarose gel electrophoresis (e.g., Fig 3, panel A lanes B and C). Conversely, amplification was not detected using either agarose gel electrophoresis or real-time fluorescence for reactions containing no DNA (e.g., Fig 3, panel A curve H and panel B lane H) or those containing total DNA from other representatives of *Phytophthora* (Table 2).

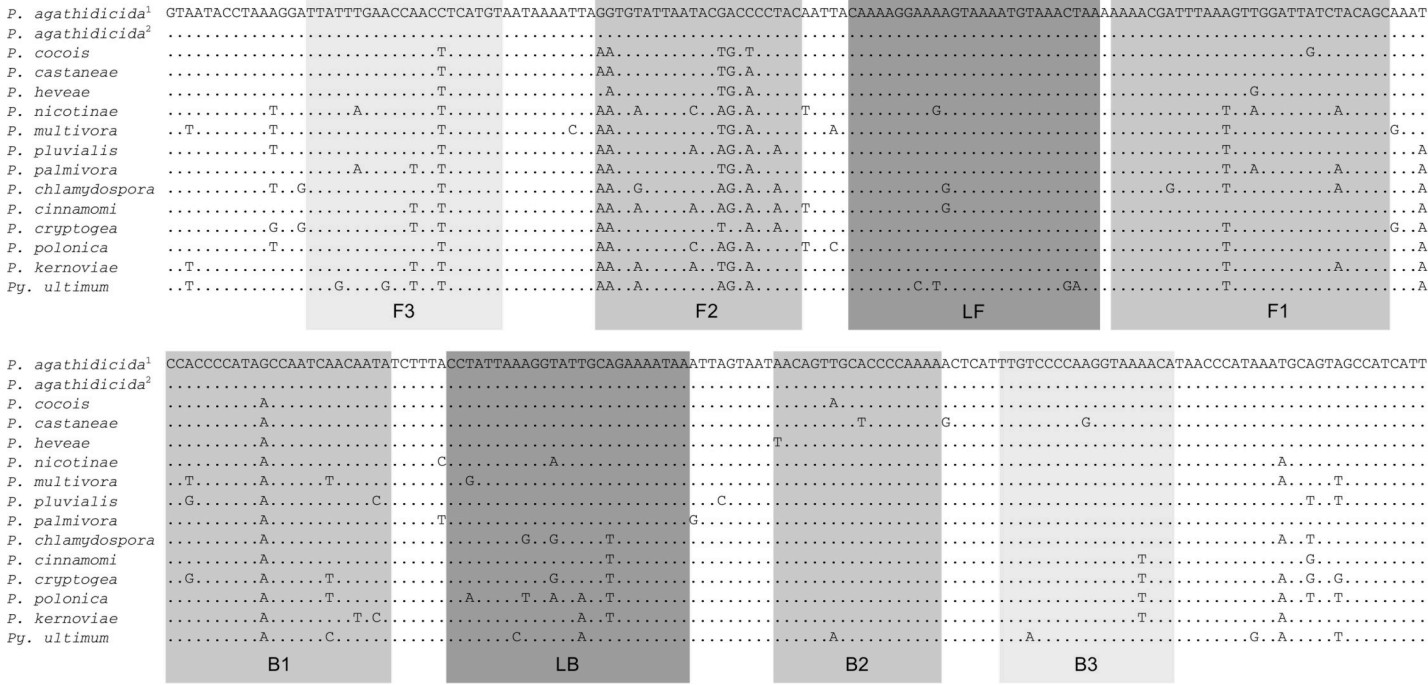

**Fig 2. Multiple sequence alignment for 13 representative *Phytophthora* species plus *Pythium ultimum* for a section of the mitochondrial genome containing the *P. agathidicida* LAMP assay target.** LAMP primer binding sites are indicated by grey outlines; F3 and B3 are binding sites for the external primer pair (i.e., PTAF3 and PTAB3), F2/F1 and B2/B1 form the binding sites for the internal primers (i.e., PTAFIP and PTABIP, respectively), and LF and LB are binding sites for the loop primers (i.e., PTALAF and PTALB).

We initially examined assay sensitivity using total *P. agathidicida* DNA. Using optimised reaction mixes and conditions, we consistently detected as little as 1 fg total DNA from cultured *P. agathidicida* isolate ICMP 18244; this limit remained the same when 2 ng total *Cedrus deodara* DNA was also added to LAMP reactions. The detection limit when using a PCR amplification product containing the target locus as template was 100 ag. Again, this limit was unchanged by the addition of 2 ng total *Cedrus deodara* DNA. Given Avogadro's number (i.e., $6.022 \times 10^{23}$ molecules/mole), the predicted length of the amplification product (i.e., 799 bp), and average weight of a base pair (i.e., 650 Daltons) the observed detection limit of 100 ag corresponds to 116 copies of the target fragment.

## Comparison of standard bioassay and hybrid LAMP bioassay

The extended and hybrid LAMP bioassays produced contrasting results for soil samples from sites in the Waitākere Ranges Regional Park and Waipoua Forest Sanctuary (Table 3). Using the extended soil bioassay, *P. agathidicida* was detected in two of six soil samples from the Waitākere Ranges Regional Park and none of the eight from the Waipoua Forest Sanctuary. Detections were five out of six from the Waitākere Ranges Regional Park and three of eight from the Waipoua Forest Sanctuary using the LAMP assay to test DNA extracted from cedar baits.

Testing of total bait DNA samples using PCR amplification and Sanger sequencing of the nrITS region was consistent with the results of the LAMP assay. Specifically, PCR amplification products of appropriate size were detected for the same five Waitākere Ranges Regional Park and three Waipoua Forest Sanctuary samples as had tested positive using the LAMP assay. Moreover, in BLAST [35] searches of the NCBI nucleotide database the DNA sequences

**Table 2. *Phytophthora* isolates used in specificity testing and results of testing with the *P. agathidicida* LAMP assay for these isolates.**

| Species and authority | Accession No. | nrITS clade | Collection location[a] | LAMP assay result |
|---|---|---|---|---|
| *Phytophthora agathidicida* B.S. Weir, Beever, Pennycook & Bellgard | ICMP 18244 | 5 | Pakiri | + |
| *P. agathidicida* | ICMP 18403 | 5 | Raetea | + |
| *P. agathidicida* | ICMP 20275 | 5 | Coromandel | + |
| *P. agathidicida* | NZFS 3118 | 5 | Huia | + |
| *P. agathidicida* | NZFS 3128 | 5 | Huia | + |
| *P. agathidicida* | NZFS 3427 | 5 | Great Barrier Island | + |
| *Phytophthora castaneae* Katsura & K. Uchida | ICMP 19450 | 5 | Lenhuachih, Taiwan | − |
| *Phytophthora cocois* B.S. Weir, Beever, Pennycook, Bellgard & J.Y. Uchida | ICMP 16949 | 5 | Kauai, Hawaii | − |
| *P. cocois* | ICMP 19685 | 5 | Port-Bouët, Cote D'Ivoire | − |
| *Phytophthora heveae* A.W. Thomps. | ICMP 19451 | 5 | Selangor, Malaysia | − |
| *Phytophthora multivora* P.M. Scott & T. Jung | ICMP 20281 | 2 | Devonport | − |
| *Phytophthora pluvialis* Reeser, Sutton & Hansen | NZFS 3563 | 3 | East Cape | − |
| *Phytophthora palmivora* (E.J. Butler) E.J. Butler | ICMP 17709 | 4 | Unknown | − |
| *Phytophthora chlamydospora* Hansen, Reeser, Sutton & Brasier | ICMP 16726 | 6 | Northland | − |
| *Phytophthora cinnamomi* Rands | ICMP 20276 | 7 | Huia | − |
| *Phytophthora cryptogea* Pethybr. & Laff. | ICMP 17531 | 8 | Mt Wellington | − |
| *Phytophthora fallax* Dobbie & M. Dick | ICMP 17563 | 9 | Katea | − |
| *Phytophthora kernoviae* Brasier | NZFS 3614 | 10 | Turitea | − |

[a]Unless otherwise indicated collection locations are within New Zealand.

of all eight amplification products shared 100% identity with 14 publically available *P. agathidicida* nrITS sequences. However, as previously reported ITS sequences do not distinguish *P. agathidicida* from *P. castaneae* [8]. Consistent with this 13 publically available sequences of *P. castaneae* were also recovered with 100% identity.

We assembled mitochondrial genome sequences from whole genome sequencing of total bait DNA from three sites in the Waitākere Ranges Regional Park. Sequences assignable to the mitochondrial genomes of *P. agathidicida*, *P. cinnamomi* and a member of the genus *Pythium* were recovered in most cases. For *P. agathidicida*, the recovered contigs corresponded to 98.6–100% of the reference mitochondrial genome; smaller portions of the *P. cinnamomi* and *Pythium* mitochondrial genomes were recovered.

## Discussion

Molecular assays have been reported for various *Phytophthora* species [e.g., 28, 41, 42]. In the present study, we have developed a LAMP assay for the detection of *P. agathidicida*, the causative agent of kauri dieback. When combined with soil baiting, this assay, which targets a region of the mitochondrial apocytochrome b gene, provides a powerful alternative to the currently used extended soil bioassay.

In specificity testing, our LAMP assay did not cross react with a range of other *Phytophthora* species, including all recognised members of Clade 5 and four of the five species (*P. nicotianae* was not tested) known to occur in kauri forest soils (Table 3). These results are consistent with pairwise comparisons of the assay target sequence from *P. agathidicida* and 64 other Oomycete taxa (S3 Table). These comparisons indicate pairwise sequence differences of 3.6–14.9% across the entire set; the remaining Clade 5 taxa (including *P.* sp. *novaeguineae*) differed by 3.5–4.4% and species from kauri forest soils (including *P. nicotianae*) by 6.1–8.3%. Together, results from specificity testing and sequence comparisons suggest our LAMP assay

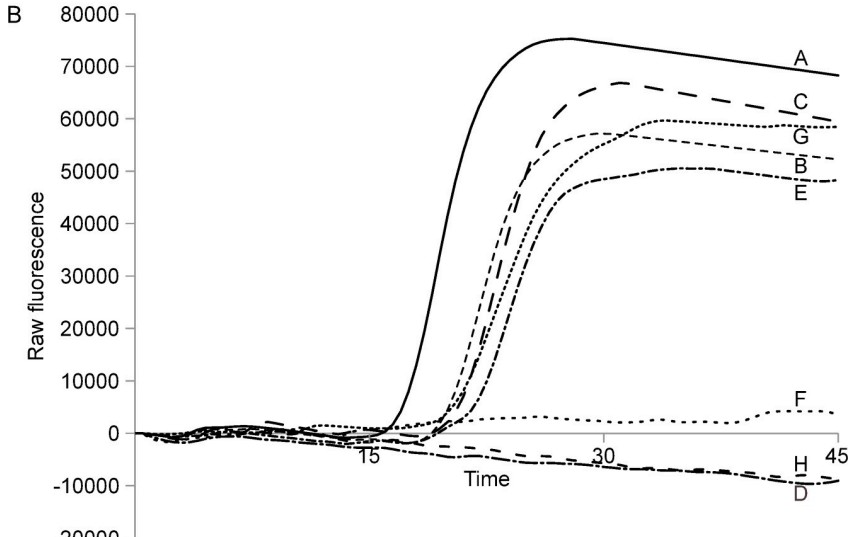

**Fig 3. Results of the *P. agathidicida* LAMP assay.** A. Endpoint visualisation of LAMP products using SYBR Safe (Invitrogen) following electrophoresis on a 1% TAE agarose gel. Lane A, 2 pg PCR amplification products from ICMP 18244; lane B, 2 pg total DNA isolate ICMP 18244; lane C, 2 pg total DNA isolate ICMP18410; lane D, 5 ng total bait DNA from Waitakere Ranges Regional Park sample HTHF 1018; lane E, 5 ng total bait DNA from Waitakere sample Ranges Regional Park HTHF 1020; lane F, 5 ng total bait DNA from Waipoua Forest Sanctuary sample HTHF 1072; lane G, 5 ng total bait DNA from Waipoua Forest Sanctuary sample HTHF 1081; lane H, no DNA control; lane L, 1 kb plus DNA ladder. B. Real-time visualisation of LAMP products using raw fluorescence data. Samples labelled as for A.

is diagnostic of *P. agathidicida*. In contrast to the currently available *P. agathidicida* RT-PCR test [15] our LAMP assay distinguishes between *P. agathidicida* and *P. castaneae*.

Our LAMP assay consistently detected 1 fg total *P. agathidicida* DNA. This detection limit is lower than that of several other *Phytophthora*-specific LAMP assays [e.g., 29] but similar to the limit reported by Than et al. [15] for their *P. agathidicida* RT-PCR assay (i.e., 2 fg). However, unlike the Than et al. [15] assay, which was ten-fold less sensitive in the presence of soil DNA, the sensitivity of our LAMP assay was found to be unchanged in the presence of background DNA. This result suggests that our LAMP assay is likely to outperform the Than et al. [15] assay for complex samples (e.g., soil or bait DNA). While it is common to report assay

**Table 3. Comparison of *Phytophthora agathidicida* detections from field collected soil samples using the extended soil bioassay and hybrid LAMP bioassay assay.**

| Accession no.[a] | Collection location | Extended soil bioassay result | Other species detected by the extended bioassay | Hybrid LAMP bioassay result |
|---|---|---|---|---|
| Waitākere Ranges Regional Park | | | | |
| HTHF 1003 | Maungaroa Ridge | – | *P. cinnamomi, Pythium senticosum* | + |
| HTHF 1014 | Maungaroa Ridge | – | – | + |
| HTHF 1018 | vicinity of Lower Kauri Track | – | *P. cinnamomi, Pythium* sp. | – |
| HTHF 1020 | vicinity of Lower Kauri Track | + | *P. cinnamomi* | + |
| HTHF 1035 | vicinity of Huia Dam | + | – | + |
| HTHF 1037 | vicinity of Huia Dam | – | *P. cinnamomi, Pythium* sp. | + |
| Waipoua Forest Sanctuary | | | | |
| HTHF 1043 | State Highway 12 | – | – | + |
| HTHF 1055 | State Highway 12 | – | *P. cinnamomi* | – |
| HTHF 1071 | State Highway 12 | – | – | – |
| HTHF 1072 | State Highway 12 | – | – | – |
| HTHF 1081 | State Highway 12 | – | *Pythium* sp. | + |
| HTHF 1083 | State Highway 12 | – | – | + |
| HTHF 1090 | Waipoua River Bridge | – | – | – |
| HTHF 1091 | Waipoua River Bridge | – | *P. cinnamomi* | – |

[a]Samples drawn from a wider set collected as part of the Healthy Trees Healthy Future programme.

detection limits in terms of the total amount of DNA, these limits can be difficult to interpret and we therefore also estimated the detection limit based on target copy number. Using a PCR-amplified target fragment the observed detection limit was approximately 116 target copies. We are not aware of estimates for the numbers of mitochondrial genomes per cell in Oomycetes but for *Saccharomyces cerevisae* 20–200 mitochondrial genome copies per cell have been reported [43]. While we acknowledge there is considerable variation in mitochondrial genome counts between taxa and life stages, the *S. cerevisae* count implies that few *P. agathidicida* zoospores need to have colonised baits before the pathogen would be detected by our LAMP assay. Indeed, the testing conducted in this study suggests that the sensitivity of our LAMP assay is sufficient to detect *P. agathidicida* at the levels typically encountered on cedar baits used for soil baiting.

For samples from both the Waitakere Ranges Regional Park and Waipoua Forest Sanctuary, *P. agathidicida* was detected with higher frequency using the hybrid LAMP bioassay than the extended soil bioassay. Specifically, four times more samples tested positive using the hybrid LAMP bioassay (Table 3). That said, results from these two approaches are consistent; those samples that tested positive using the extended soil bioassay also tested positive using the hybrid LAMP bioassay. Given these markedly different results we used a PCR-based approach to further assess the presence of *P. agathidicida*. Amplification products that shared 100% identity with *P. agathidicida* ITS sequences were recovered for all eight samples that had tested positive using the hybrid LAMP bioassay. Although these ITS sequences are also consistent with the presence of *P. castaneae* [8], this latter species has not been reported from New Zealand suggesting *P. agathidicida* is the likely source. Moreover, from whole genome sequencing of total bait DNA we recovered complete, or nearly so (i.e., 98.6–100%), mitochondrial genome sequences from each of the three Waitakere Ranges Regional Park samples we examined. In all cases the recovered mitochondrial genome included an intact version of the LAMP assay target sequence. For two of these samples (i.e., HTHF 1020 and HTHF 1035) both the extended and hybrid LAMP bioassays had detected the presence of *P. agathidicida* whereas for the remaining sample (i.e., HTHF 1014) *P. agathidicida* had only been detected using the

hybrid LAMP bioassay. Taken together these additional analyses strongly support the results of the hybrid LAMP bioassay; that is, that *P. agathidicida* was present in these samples. More generally our analyses also imply that although *P. agathidicida* zoospores may colonise plant tissue baits, this will not always result in visual detection of *P. agathidicida* following culturing.

At least partially the slow average *in vitro* growth rate of *P. agathidicida* (4.5 mm/day; [8]) may explain why culturing resulted in fewer *P. agathidicida* detections than did genetic testing. Specifically, for two thirds of the samples (i.e., four of the six) where the results of the extended and hybrid LAMP bioassay differed, oomycetes with faster *in vitro* growth rates–specifically, *P. cinnamomi* (8.3 mm/day [44]) and *Pythium* sp. (21–29 mm/day [45])–were found on the plates following culturing (Table 3). For these four samples our ability to visually detect *P. agathidicida* may have been compromised by the presence of faster growing species. If so, stochastic differences in patterns of competition across replicates may explain the inconsistent recovery of *P. agathidicida* from split replicate soil samples [46, 47]. That said, in these studies the choice of bait tissue was not standardised. Recently, Khaliq et al. [48] have shown that bait type and integrity (e.g., detached or intact) influences the diversity of *Phytophthora* recovered by traditional baiting and culturing. There are clearly multiple factors that impact upon our ability to detect *P. agathidicida*; at least some of these are either reduced or eliminated by the use of a genetic test to evaluate presence of the pathogen.

Our LAMP assay could be applied to DNA from other sources. One possibility would be to directly test soil DNA for the presence of *P. agathidicida*. However, low pathogen titre and soil heterogeneity pose considerable challenges for this approach; indeed, inconsistencies between RT-PCR testing of soil DNA and the extended bioassay have previously been reported [47]. Another possibility would be to test DNA from diseased tissue. In this case testing would be confirmatory and not provide information about the distribution of the pathogen beyond those sites where physical symptoms have already been recognised. Given these limitations we have instead focused on implementing a hybrid bioassay that combines baiting as a means of minimising the impact of soil heterogeneity and low pathogen titre with a LAMP assay that increases the sensitivity and reproducibility of detection from baits. Additionally by removing the need for culturing and morphological identification, as well as confirmatory sub-culturing and RT-PCR testing, this hybrid bioassay is both faster and more cost effective than the current extended soil bioassay. As a result the hybrid LAMP bioassay could dramatically enhance our ability to address the threat of kauri dieback. In particular, given the cost effectiveness of this diagnostic we could move from confirmatory testing at diseased sites to systematic monitoring of the pathogen across the distribution of kauri. The latter is necessary if we are to determine pathogen distribution, measure the rate and pattern of spread, and evaluate the efficacy of disease control measures.

Additionally, since culturing is not required, the hybrid LAMP bioassay can be performed without centralised laboratory facilities. Although we acknowledge that the approach is not equipment free, devices for DNA extraction and amplification are now available that enable testing to be carried out locally [e.g., 49, 50, 51]. Critically, the ability to implement testing outside a laboratory creates opportunities for landowners and community groups to engage directly with diagnostic technologies and hence with disease management and conservation programmes. The information provided by community-led testing could enhance the management of local kauri stands as well as contribute directly to regional and national initiatives.

## Supporting information

**S1 Table. Sources of the oomycete mitochondrial genome sequences used when designing the *P. agathidicida* loop-mediated isothermal amplification (LAMP) assay.**
(DOCX)

**S2 Table. Location details for field collected soil samples used in comparisons of extended bioassay and *P. agathidicida* loop-mediated isothermal assay.**
(DOCX)

**S3 Table. Comparison of the *Phytophthora agathidicida* LAMP assay target sequence and the corresponding locus in other oomycetes.**
(DOCX)

**S1 Fig. Endpoint visualisation of PCR amplification products using SYBR safe (Invitrogen) following electrophoresis on a 1% TAE agarose gel.**
(PDF)

**S1 Raw images. Original images supporting Fig 3 and S1 Fig.**
(PDF)

## Acknowledgments

We acknowledge the mana whenua Te Kawerau ā Maki (Waitākere Ranges) and Te Roroa (Waipoua). We thank the International Collection of Microorganisms from Plants (ICMP) and New Zealand Forest Research Institute (NZFS) culture collections as well as the HTHF programme for access to samples.

## Author Contributions

**Conceptualization:** Richard C. Winkworth, Peter J. Lockhart.

**Formal analysis:** Richard C. Winkworth.

**Investigation:** Richard C. Winkworth, Briana C. W. Nelson, Stanley E. Bellgard, Chantal M. Probst, Patricia A. McLenachan.

**Methodology:** Richard C. Winkworth, Stanley E. Bellgard.

**Project administration:** Richard C. Winkworth.

**Resources:** Stanley E. Bellgard, Chantal M. Probst.

**Validation:** Richard C. Winkworth, Briana C. W. Nelson.

**Visualization:** Richard C. Winkworth.

**Writing – original draft:** Richard C. Winkworth, Briana C. W. Nelson.

**Writing – review & editing:** Richard C. Winkworth, Briana C. W. Nelson, Stanley E. Bellgard, Chantal M. Probst, Patricia A. McLenachan, Peter J. Lockhart.

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
