## [Decision Letter · Decision Letter 0]

29 Oct 2019

PONE-D-19-27464

A LAMP at the end of the tunnel: a rapid, field deployable assay for the kauri dieback pathogen, Phytophthora agathidicida

PLOS ONE

Dear Dr Winkworth,

Thank you for submitting your manuscript to PLOS ONE. After careful consideration, we feel that it has merit but does not fully meet PLOS ONE’s publication criteria as it currently stands. Therefore, we invite you to submit a revised version of the manuscript that addresses the points raised during the review process.

I ask the authors of this work to respond to the comments of reviewers, with whom I completely agree.

We would appreciate receiving your revised manuscript by Dec 13 2019 11:59PM. To enhance the reproducibility of your results, we recommend that if applicable you deposit your laboratory protocols in protocols.io, where a protocol can be assigned its own identifier (DOI) such that it can be cited independently in the future. For instructions see: http://journals.plos.org/plosone/s/submission-guidelines#loc-laboratory-protocols

We look forward to receiving your revised manuscript.

Kind regards,

Ruslan Kalendar, PhD

Academic Editor

PLOS ONE

Journal Requirements:

We also acknowledge financial support from the BioProtection Research Centre (P.J.L. and R.C.W.), Massey University (R.C.W. and P.J.L.), and the New Zealand Ministry of Business, Innovation and Employment via the Catalyst: Seeding Fund (administered by the Royal Society Te Apārangi; P.J.L. and R.C.W) and the Strategic Science Investment Fund (S.E.B).

P.J.L. and R.C.W., BPRC_MU_2016_1, BioProtection Research Centre (bioprotection.org.nz). The funders had no role in study design, data collection and analysis, decision to publish, or preparation of the manuscript.

P.J.L. and R.C.W., MAU1702, New Zealand Ministry of Business, Innovation and Employment Catalyst: Seeding Fund (https://royalsociety.org.nz/what-we-do/funds-and-opportunities/catalyst-fund/catalyst-seeding/). The funders had no role in study design, data collection and analysis, decision to publish, or preparation of the manuscript.

R.C.W. and P.J.L., NP94607, Massey University (massey.ac.nz). The funders had no role in study design, data collection and analysis, decision to publish, or preparation of the manuscript.

S.E.B, SIF 228001 0032, New Zealand Ministry of Business, Innovation and Employment Strategic Science Investment Fund (https://www.mbie.govt.nz/science-and-technology/science-and-innovation/funding-information-and-opportunities/investment-funds/strategic-science-investment-fund/). The funders had no role in study design, data collection and analysis, decision to publish, or preparation of the manuscript.

Reviewers' comments:

Reviewer's Responses to Questions

**Comments to the Author**

1. Is the manuscript technically sound, and do the data support the conclusions?

Reviewer #1: Yes

Reviewer #2: Partly

2. Has the statistical analysis been performed appropriately and rigorously? 

Reviewer #1: N/A

Reviewer #2: N/A

3. Have the authors made all data underlying the findings in their manuscript fully available?

Reviewer #1: Yes

Reviewer #2: Yes

4. Is the manuscript presented in an intelligible fashion and written in standard English?

Reviewer #1: Yes

Reviewer #2: Yes

5. Review Comments to the Author

Reviewer #1:

I am persuaded that the authors have developed a LAMP assay for Phytophthora agathidicida that is more sensitive and probably more accurate than the current morphological assessment to detect this pathogen. It is faster – probably cutting in half the total time to complete a diagnosis.

The claims that the LAMP approach is better are accurate, but biased. If the authors mention the lack of a need for centralized lab facility for the traditional approach, they also need to mention the need for materials and equipment to conduct the LAMP assay. I also believe that the suggestion that local communities will be empowered to conduct their own assays is a bit overstated. Finally, monitoring is mentioned. When monitoring, there is a need to develop a reasonable sampling plan. Sampling near symptomatic plants is now highly biased for success. Just because there is a better detection from baits does not automatically mean that general monitoring will become useful.

In several locations the authors mention that one of six samples in the Waitakere Ranges National Park was positive for P. agathidicida using the traditional assay. However, my version of Table 2 indicates two positives were detected by the traditional assay. This discrepancy needs to be clarified.

To me it seems very important that each of the “traditional” positives was also positive based on the LAMP assay. Despite the tiny sample size, I think the authors could make more of that consistency.

Reviewer #2:

I really enjoyed the whole approach employed by the authors to develop the LAMP assay, in particular the work done to select the appropriate locus was quite impressive. I also enjoyed reading the manuscript: excellent style made it fairly straightforward to follow. I do however have some comments that may improve the paper.

1)- This is more of a question than a comment, but I suppose that if inoculum is found in the soil and the pathogen is baited from soil, we may actually be looking at a root pathogen as well as a root collar pathogen. Either way, please explain or amend the text accordingly

2)- I am very familiar with the baiting approach, and I do believe it is an effective way to detect inoculum in soil, thanks to motility of zoospores. However I am surprised the authors did not test the technique directly on soil and on symptomatic plant tissue. I strongly recommend these two tests to be included in a revised version

3)- I think it may be important to confirm the bait-negative but LAMP positive samples are indeed infested by P. agathicida. I recommend using the available PCR assay or direct sequencing on such samples to make sure the interpretation of the results is correct

4)- Finally is it possible to PCR amplify the LAMP positive samples and sequence them to confirm the positives are so because the target really is there? I can live with positives from baiting as support of the LAMP results, but this comment even strengthens the need to deal with my comment 3 above.

Congratulations to the authors for a nice study

6. PLOS authors have the option to publish the peer review history of their article (what does this mean?). If published, this will include your full peer review and any attached files.

Reviewer #1: No

Reviewer #2: No

---

## [Author Response · Author response to Decision Letter 0]

19 Dec 2019

5. Review Comments to the Author

Reviewer #1:

I am persuaded that the authors have developed a LAMP assay for Phytophthora agathidicida that is more sensitive and probably more accurate than the current morphological assessment to detect this pathogen. It is faster – probably cutting in half the total time to complete a diagnosis.

The claims that the LAMP approach is better are accurate, but biased. If the authors mention the lack of a need for centralized lab facility for the traditional approach, they also need to mention the need for materials and equipment to conduct the LAMP assay. I also believe that the suggestion that local communities will be empowered to conduct their own assays is a bit overstated. Finally, monitoring is mentioned. When monitoring, there is a need to develop a reasonable sampling plan. Sampling near symptomatic plants is now highly biased for success. Just because there is a better detection from baits does not automatically mean that general monitoring will become useful.

We acknowledge this portion of the manuscript was not sufficiently well developed.

One of the advantages of the LAMP technology is that it is well suited to use outside a laboratory situation (noted on page 4); indeed there is ample literature describing the use of this technology in “field” settings. As requested we have added a statement mentioning the requirement for protocols or devices that would enable the LAMP technology to be used outside of a laboratory on page 22.

We have modified discussion of use of our test by local communities. While developing a test that would empower local communities was an aim of the project we do not directly address this potential here. That said, we currently have a project involving use of the test by a local community group so wish to highlight this as a possibility.

In terms of monitoring we acknowledge that current testing is limited to locations where symptomatic plants have been identified; this has led to gaps in our understanding especially around pathogen distribution and spread. However, we think that the proposed hybrid assay addresses this limitation not because it offers more robust detection of the pathogen but because it can be conducted faster and much more cheaply (on the order of 10 times based on charge out rates we are aware of) than the extended soil bioassay. The difference is such that systematic sampling of kauri forests becomes a possibility. That is, use of the hybrid LAMP bioassay would allow us to move from targeted sampling of symptomatic sites to collection and testing based on sampling plans specifically designed to address specific research questions. We now more fully describe our reasoning for suggesting that the hybrid LAMP bioassay could enable widespread monitor on page 22.

In several locations the authors mention that one of six samples in the Waitakere Ranges National Park was positive for P. agathidicida using the traditional assay. However, my version of Table 2 indicates two positives were detected by the traditional assay. This discrepancy needs to be clarified.

Sorry this was a error carried over from an earlier draft of the manuscript. As noted in Table 2 there were two Waitakere Ranges Regional Park samples that tested positive using the extended bioassay. This has been corrected in the text.

To me it seems very important that each of the “traditional” positives was also positive based on the LAMP assay. Despite the tiny sample size, I think the authors could make more of that consistency.

Thanks for this suggestion we now make this point on page 20.

Reviewer #2:

I really enjoyed the whole approach employed by the authors to develop the LAMP assay, in particular the work done to select the appropriate locus was quite impressive. I also enjoyed reading the manuscript: excellent style made it fairly straightforward to follow. I do however have some comments that may improve the paper.

1)- This is more of a question than a comment, but I suppose that if inoculum is found in the soil and the pathogen is baited from soil, we may actually be looking at a root pathogen as well as a root collar pathogen. Either way, please explain or amend the text accordingly

This has been ammended as requested in both the abstract and introduction.

2)- I am very familiar with the baiting approach, and I do believe it is an effective way to detect inoculum in soil, thanks to motility of zoospores. However I am surprised the authors did not test the technique directly on soil and on symptomatic plant tissue. I strongly recommend these two tests to be included in a revised version.

We agree that baiting is an effective way of recovering inoculum from soil samples. That said we also think that using a molecular diagnostic, rather than culturing and visual detection, extends the effectiveness of baiting. 

We had not pursued direct testing of soils or of diseased tissues; these sample types have technical limiations and/or would not allow us to move beyond confirmatory testing. Moreover, cultural and regulatory constraints mean that there are long lead in times to obtain samples. Given the relatively short turn around time for this revision we have therefore been unable to explore these sample types.

We now breifly dicussion why we focused on development of a hybrid bioassay on pages 22. We hope that this addresses the point sufficiently. 

3)- I think it may be important to confirm the bait-negative but LAMP positive samples are indeed infested by P. agathicida. I recommend using the available PCR assay or direct sequencing on such samples to make sure the interpretation of the results is correct.

4)- Finally is it possible to PCR amplify the LAMP positive samples and sequence them to confirm the positives are so because the target really is there? I can live with positives from baiting as support of the LAMP results, but this comment even strengthens the need to deal with my comment 3 above.

Congratulations to the authors for a nice study

Comments 3 and 4 are linked, both requesting additional work to support positive LAMP results when the extended bioassay had not detected P. agathidicida for the same samples.

In the original version of the manuscript we reported WGS of total bait DNA from several of the HTHF samples that had tested positive using our LAMP assay. More than 95% of the P. agathidicida mitochondrial genome, including the LAMP target sequence, was recovered in all cases. We have revised the text to better make this point. In particular we now refer to sequencing of three samples two where both the hybrid LAMP and extended bioassays had been positive for P. agathidicida and one where only the hybrid LAMP had been positive.

As suggested we have also added a description of testing based on PCR amplification and sequencing of the nuclear ribosomal ITS. For this we have been unable to use the available RT-PCR assay (Than et al., 2013). Instead we combined one of Than et al.’s (2013) P. agathidicida-specific primers with one of White et al.’s (1990) generic primers in order to generate a larger PCR product more suited to DNA sequence analyses. Our PCR testing is entirely consistent with the LAMP testing. That is, we only recover PCR amplification products from those samples that had previously tested positive using our LAMP assay and DNA sequences from those fragments share 100% identity with previously published P. agathidicida ITS sequences.

We hope that this addresses these points sufficiently.

---

## [Editor Report · Decision Letter 1]

23 Dec 2019

A LAMP at the end of the tunnel: a rapid, field deployable assay for the kauri dieback pathogen, *Phytophthora agathidicida*

PONE-D-19-27464R1

Dear Dr. Winkworth,

We are pleased to inform you that your manuscript has been judged scientifically suitable for publication and will be formally accepted for publication once it complies with all outstanding technical requirements.

With kind regards,

Ruslan Kalendar, PhD

Academic Editor

PLOS ONE

---

## [Editor Report · Acceptance letter]

8 Jan 2020

PONE-D-19-27464R1 

A LAMP at the end of the tunnel: a rapid, field deployable assay for the kauri dieback pathogen, *Phytophthora agathidicida*

Dear Dr. Winkworth:

I am pleased to inform you that your manuscript has been deemed suitable for publication in PLOS ONE. Congratulations! Your manuscript is now with our production department. 

With kind regards,

on behalf of

Dr. Ruslan Kalendar 

Academic Editor

PLOS ONE